# Understanding the Role of Alcohol in Metabolic Dysfunction and Male Infertility

**DOI:** 10.3390/metabo14110626

**Published:** 2024-11-15

**Authors:** Valentina Annamaria Genchi, Angelo Cignarelli, Andrea Sansone, Dimitri Yannas, Leonardo Dalla Valentina, Daniele Renda Livraghi, Giorgia Spaggiari, Daniele Santi

**Affiliations:** 1Department of Precision and Regenerative Medicine and Ionian Area-Section of Internal Medicine, Endocrinology, Andrology and Metabolic Diseases, University of Bari Aldo Moro, 70121 Bari, Italy; 2Chair of Endocrinology and Medical Sexology (ENDOSEX), Department of Systems Medicine, University of Rome Tor Vergata, Tower E South, Room E 413, Via Montpellier 1, 00133 Rome, Italy; 3Department of Biomedical, Metabolic and Neural Sciences, University of Modena and Reggio Emilia, 41121 Modena, Italydaniele.santi@unimore.it (D.S.); 4Unit of Endocrinology, Department of Medical Specialties, Azienda Ospedaliero-Universitaria of Modena, 41125 Modena, Italy

**Keywords:** alcohol, testosterone, hypogonadism, infertility

## Abstract

*Purpose:* Over the past 40–50 years, demographic shifts and the obesity epidemic have coincided with significant changes in lifestyle habits, including a rise in excessive alcohol consumption. This increase in alcohol intake is a major public health concern due to its far-reaching effects on human health, particularly on metabolic processes and male reproductive function. This narrative review focuses on the role of alcohol consumption in altering metabolism and impairing testicular function, emphasizing the potential damage associated with both acute and chronic alcohol intake. *Conclusion:* Chronic alcohol consumption has been shown to disrupt liver function, impair lipid metabolism, and dysregulate blood glucose levels, contributing to the development of obesity, metabolic syndrome, and related systemic diseases. In terms of male reproductive health, alcohol can significantly affect testicular function by lowering testosterone levels, reducing sperm quality, and impairing overall fertility. The extent of these effects varies, depending on the frequency, duration, and intensity of alcohol use, with chronic and abusive consumption posing greater risks. The complexity of alcohol’s impact is further compounded by individual variability and the interaction with other lifestyle factors such as diet, stress, and physical activity. Despite growing concern, research on alcohol’s effects remains inconclusive, with significant discrepancies across studies regarding the definition and reporting of alcohol consumption. These inconsistencies highlight the need for more rigorous, methodologically sound research to better understand how alcohol consumption influences metabolic and reproductive health. Ultimately, a clearer understanding is essential for developing targeted public health interventions, particularly in light of rising alcohol use, demographic changes, and the ongoing obesity crisis.

## 1. Introduction

Alcohol addiction represents a global health morbidly issue, associated with increased deaths (5–8%) and an overall higher risk of developing important metabolic morbidities. Alcohol is a toxic, psychoactive, and dependence-producing substance and the World Health Organization has established its role at contributing to psychological and social issues. In the literature, long-term alcohol consumption is related to over 200 pathological conditions, including at least seven types of cancer. In particular, alcohol abuse can impair several organs, such as the neurological system [1], endocrine system [2], liver [3], cardiovascular system [4], and gastrointestinal tract [1], and the metabolism of vital nutrients [5].

The aim of this review is to provide an updated overview of the potential effects of alcohol consumption on the gonadal axis in men. To understand the consequence of alcohol consumption on gonadal function, the physiological and pathophysiological mechanisms involved in alcohol metabolism are synthesized herein.

## 2. Alcohol and Metabolic Health

### 2.1. The Physiology of Alcohol Metabolism

Alcohol is absorbed from the stomach to the small intestine, entering the bloodstream after 30 to 45 min from ingestion through passive diffusion. Next, alcohol reaches the liver where it is metabolized by specific enzymes. The rate of blood alcohol concentration increase strongly depends on the grade of gastric emptying and on the first-pass liver metabolism. Moreover, environmental factors, such as the frequency and type of alcohol consumption, genetic factors, and different expression of two main alcohol-metabolizing enzymes, (alcohol dehydrogenase [ADH] and aldehyde dehydrogenase [ALDH2]) could also affect the blood alcohol concentration. In addition, alcohol elimination shows individual differences, influenced by age, smoking, nutrition, chronic alcohol usage, and time of day [6,7].

Physiologically, in the liver, alcohol metabolism occurs through the activation of both oxidative and non-oxidative pathways. Three main pathways metabolize alcohol in hepatocytes through the oxidative mechanism. First, cytosolic ADH oxidizes ethanol into acetaldehyde, a toxic molecule able to bind proteins, DNA, or lipids, impairing their function, as we will discuss later. Alcohol oxidation by ADH involves multiple steps: (i) the ADH enzyme binds to the NAD^+^ cofactor, (ii) an alcohol substrate displaces a water molecule coordinated to Zn^2+^, (iii) the bound alcohol is deprotonated, forming a zinc alkoxide intermediate, (iv) a hydride ion is transferred from the alcohol to the NAD+ cofactor, producing a Zn^2+^-bound aldehyde product and NADH, (v) the aldehyde product is replaced by a water molecule to restore the original Zn^2+^ catalytic site, and (vi) NADH is released from the enzyme, completing the catalytic cycle [8].

Second, acetaldehyde is also produced after ethanol oxidation by a microsomal ethanol oxidizing system (MEOS) composed by several cytochromes, including P450 2E1 (CYP2E1), producing reactive oxygen species (ROS) and leading to oxidative stress and inflammation [9]. Third, acetaldehyde is converted into acetate by ALDH, localized in the mitochondria of the hepatocyte, which increases liver blood flow, suppresses the central nervous system, and disrupts various metabolic processes [10]. In addition, acetate can be converted into acetyl-CoA, a crucial molecule for lipid and cholesterol synthesis within the mitochondria. Depending on the body’s nutritional, energy, and hormonal status, acetyl-CoA can then be further processed into CO_2_, fatty acids, ketones, or cholesterol [8].

The enzymes ADH and ALDH have multiple forms, known as isoenzymes, deriving from genetic variations, characterized by a different enzymatic activity which in turn affects the individual alcohol metabolism. Typically, alcohol is broken down at a rate of 80–130 mg per kilogram of body weight per hour, or approximately 6–10 g per hour. The expression of specific isoenzymes within the stomach and liver also shows an inter-individual variability, thus leading to differences in metabolic efficiency.

The non-oxidative pathways are responsible of the metabolism of a minor portion of the alcohol. This step consists of the non-oxidative conjugation of alcohol with various endogenous metabolites through the action of specific enzymes. For instance, esterification of alcohol with fatty acids (FAs) produces fatty acid ethyl esters (FAEEs), while phospholipase D (PLD) catalyzes the transphosphatidylation of phosphatidylcholine with ethanol, resulting in phosphatidylethanol (PEth). Additionally, alcohol conjugation with glucuronic acid and sulfate forms ethyl glucuronide (EtG) and ethyl sulfate (EtS), respectively [11,12,13].

### 2.2. Pathophysiology of Alcohol-Associated Diseases

Mild-to-moderate drinking habits are related to the development of intermediate phenotypes of metabolic syndrome, including type 2 diabetes mellitus [14], arterial hypertension [15], central obesity, liver steatosis, cardiovascular disease, and systemic inflammation [16], suggesting that multiple pathogenetic mechanisms often act simultaneously in the development of full-blown metabolic disease. The severity of the pathological condition of alcohol assumption seems to be dose-dependent. Accordingly, alcohol abuse is known to increase the risk of multiorgan damage through inflammation, enhanced oxidative stress, aberrant post-translational protein modifications, and impaired lipid metabolism as well [17]. However, inter-individual variability in susceptibility to alcoholic liver disease exists independent of the daily amount of alcohol consumption, highlighting the existence of multiple mechanisms involved in liver and metabolic dysfunction. After consuming alcohol, ethanol impairs mouth physiology, increasing oral mucosa permeability and sensitivity to other toxic substances [18]. Moreover, ethanol increases the permeability of the gut barrier, enhancing local and systemic inflammation and microbial dysbiosis.

The liver is the first organ where alcohol exerts detrimental effects, since it is mainly metabolized by hepatic cells expressing high levels of key enzymes involved in the oxidation and dehydration of alcohol. The harmful effects of alcohol on liver function have been observed in several population-based studies where the presence of metabolic syndrome and weekly binge drinking had substantial supra-additive effects on liver-related outcomes, increasing the risk of all-cause mortality [19].

Considering the histopathology of the liver in alcohol abusers, different alcohol-associated liver diseases (ALDs) have been differentiated ranging from asymptomatic to severe forms with higher mortality. ALD progression may include stages from steatosis to steatohepatitis and fibrosis, potentially leading to cirrhosis and related complications, and, in some cases, hepatocellular carcinoma [20]. Among these conditions, fatty liver disease (ALF) represents a reversible liver injury upon alcohol abstinence without fatal outcomes, excluding in severe cases, characterized by a fat deposition in over 5% of hepatocytes, presenting primarily as macrovesicular steatosis, with microvesicular forms being rare [21]. Otherwise, alcoholic steatosis hepatitis (ASH) is characterized by steatosis and liver damage consisting into swollen hepatocytes with rarified cytoplasm, clumping of intermediate filaments, and loss of staining for cytokeratin 8 and 18 together with low level necrotic or apoptotic alterations [22]. Alcoholic hepatitis (AH) is the main severe form of ALD with pronounced liver necrosis and apoptosis as compared to ASH in association with infiltration of inflammatory cells [22]. Patients with a history of heavy daily alcohol use (>40 g for women and 60 g for men) for at least six months frequently receive a diagnosis of AH in terms of the sudden onset of jaundice, liver failure symptoms (ascites, hepatic encephalopathy, or upper gastrointestinal bleeding) in association with anorexia, fever, and abdominal pain.

Alcohol-induced liver damage is also associated with the onset of metabolic dysfunction including type 2 diabetes (T2D), whose risk of development changes according to the amount and frequency of alcohol daily intake. Light and moderate alcoholics show a lower risk of T2D, whereas heavy alcohol abusers do not show a increased risk of developing T2D [14,23,24]. In accordance, multiple meta-analyses have confirmed a link between alcohol consumption and the incidence of T2D, observing that the consumption of 24–48 g of alcohol per day is associated with an approximately 20–30% reduction in risk [23,25].

Importantly, alcohol intake exerts a stressful effect on insulin-sensitive tissues since the CYPE21-mediated oxidative stress experienced during alcoholic liver injury impairs insulin signaling. In fact, chronic ethanol-induced hepatic steatosis was associated with a decreased phosphorylation of Akt at Thr308 in the livers of mice, together with the increased generation of adducts of ethanol-derived metabolites and Akt, thus promoting insulin resistance [26].

In addition, a growing body of evidence has reported the harmful effect of alcohol intake on insulin sensitivity, observing that chronic ethanol exposure reduces insulin receptor binding and inhibition of downstream signaling via the Ras/Raf/Erk MAPK pathway, which reduces DNA synthesis and liver regeneration in a PI3K–Akt-dependent manner, thus promoting cell death, energy metabolism, and mitochondrial function [27]. In addition, the presence of insulin resistance during alcohol overload participates in the aggravation of liver dysfunction as observed in ethanol-fed mice where insulin insensitivity exacerbates apoptosis and mitochondrial abnormalities due to increased oxidative stress and decrease the antioxidant defense via suppressing the Nrf-2 pathway [28]. Detrimental effects of alcohol were also verified in mice with moderate obesity, where the chronic administration of alcohol led to a significant body weight gain and glucose intolerance together with an increase in steatohepatitis and liver fibrosis, effects caused by dysregulated leptin/(5′ AMP-activated protein kinase) AMPK signaling [29]. Although corroborating results have demonstrated that metabolic syndrome and alcohol consumption act synergistically in favoring liver illness, nowadays the mechanisms underlying alcohol-induced metabolic dysfunction still remain elusive. However, oxidative stress, defects in mitochondria function, impaired lipid metabolism, and inflammation in concert play a fundamental role in promoting dysmetabolism under alcohol abuse.

#### 2.2.1. Oxidative Stress

As reported above, alcohol metabolism is mostly driven by ADH and ALDH enzymes which favor the alterations of cellular redox state, hypoxia, formation of acetaldehyde, depletion of antioxidants, mitochondrial damage, and activation of Kupffer cells [30]. Several studies have observed that oxidative ethanol-derived metabolites increase oxidative stress due to both dehydrogenases and CYP2E1 enzyme activity; this in turn raises the levels of NADH and its following oxidation by xanthine oxidase, leading to ROS formation [31].

In this setting, acetaldehyde is the main contributor of increased oxidative status under alcohol overload because it participates in the impairment of the cellular redox state via multiple mechanisms [32]: (i) it favors an intracellular redox imbalance by raising levels of oxidative stress markers, such as malondialdehyde (MDA), and by decreasing the concentration of the antioxidant glutathione (GSH) [33]; (ii) it reduces the enzymatic activity of superoxide dismutase 2, a potent endogenous antioxidant [34]; (iii) it increases NADPH oxidase’s expression, enzymes contributing to the phagocyte-mediated host defense, and its regulatory proteins (i.e., p22phox, p47phox, and p67phox) resulting in enhanced production of O_2_^−^ and H_2_O_2_ and lower GSH levels [35].

The mitochondrial respiratory chain is also involved in the overproduction of ROS due to chronic alcohol consumption. High levels of reducing equivalents, such as NADH generated by ADH activity, enter the mitochondrial respiratory chain leading to an over-reduction of the electron transport chain and promoting the formation of superoxide anions [36].

The abundance of ethanol-induced free radicals triggers a process called lipid peroxidation, in which various byproducts of alcohol breakdown, as adducts, are generated [37]. Adducts are formed when acetaldehyde or other aldehydes bind with proteins. Notable forms of adduct include the aldehydes MDA and 4-hydroxy-2-nonenal (HNE). Similar to acetaldehyde, these reactive aldehydes are not able to cross the blood–brain barrier. In the presence of acetaldehyde, MDA can interact with proteins to create malondialdehyde–acetaldehyde (MAA) adducts, which are known to exhibit proinflammatory properties [38].

In addition, ROS upregulate multiple transcription factors including heat shock factors and sterol regulatory element binding protein 1c/2 (SREBP-1c/2) which in turn lead to lipid accumulation, a crucial step in the development of alcoholic liver disease [39] (Figure 1). Indeed, the accumulation of intrahepatic lipids represents the first step after excessive alcohol addiction due to an imbalance between triglycerides synthesis and oxidation which occurs for the raised levels of NADH after ethanol oxidation by ADH which subsequently impaired the β-oxidation of lipids [40]. This mechanism was further elucidated by experiments in mice with a deficiency of the farnesoid X receptor (FXR), a member of the nuclear receptor superfamily of transcription factors more highly expressed in liver and able to regulate bile acids synthesis, the key hepatic product facilitating the solubilization and absorption of cholesterol, dietary lipids, and fat-soluble vitamins in the intestines. Mice depleted in FXR exhibited more severe liver injury, steatosis, inflammation and fibrosis after alcohol consumption compared to control mice, suggesting that the lack of expression of this receptor exacerbates alcoholic liver disease [41]. Accordingly, when hepatic cells lacking FXR were exposed to ethanol, an increase in ROS generation was observed, suggesting that the ROS increase is essential in accelerating liver damage [41].

#### 2.2.2. Lipid Metabolism

The impact of alcohol abuse on lipid metabolism was well confirmed by a recent meta-analysis including 26 trials whose results showed that beer drinkers showed significantly higher total cholesterol (3.52 mg/dL), HDL cholesterol (3.63 mg/dL), and apolipoprotein A1 (0.16 mg/dL) as compared to controls. However, no significant differences were found in LDL cholesterol (−2.85 mg/dL) or triglyceride levels (0.40 mg/dL) [42]. Although these results indicate a potential beneficial effect of alcohol on lipid metabolism, in vivo findings reported a wide range of effects of ethanol-derived metabolites on this endpoint. For instance, acetaldehyde participates in liver damage by activating lipolysis in adipose tissue and upregulates the expression of FA transporter proteins (FATPs) and FA translocase (FAT/CD36) in the liver which in turn raise FA uptake by hepatocytes [43,44]. Furthermore, acetaldehyde inhibits the AMPK pathway [45], favors the increase in lipid synthesis through the upregulation of lipin 1-β [46] and dysregulates mitochondrial β-oxidation [47,48] (Figure 1). Interestingly, the detrimental effect of ethanol on liver function has been well appreciated in a mice model of acute alcoholic liver injury with ADH1 deficiency. In this model, several dysregulated genes were identified in the livers after ethanol gavage (7 g/kg) in terms of the upregulation of genes related to de novo lipogenesis (*Acaca*, *Fasn*, and *Cyp51*) in association with enhanced levels of serum and specific hepatic FFAs (palmitic, oleic, and linoleic acids), increased adipocyte death, and lipolysis [49]. FA catabolism is also impaired by alcohol-derived acetaldehyde via the inactivation of peroxisome proliferator-activated receptor alpha (PPARα), a transcriptional factor regulating the expression of enzymes involved in FA oxidation in the mitochondria including carnitine palmitoyltransferase 1 (CPT1) [50]. Other mechanisms detected to be associated with the liver steatosis pathogenesis are autophagy and lysosome functions, which are impaired under alcohol consumption in a time-dependent manner. Indeed, long-term alcohol exposure inhibits autophagy through the suppression of the AMPK pathway. Conversely, short-term ethanol exposure activates autophagy by generating acetaldehyde and ROS and inhibiting mTOR, suggesting that the acute activation of autophagy is a compensatory mechanism preventing the onset of steatosis during the early stages of alcoholic liver injury [51]. Furthermore, the impairment of lipid metabolism under ethanol abuse is also due to an increase in acetate production associated with a raise in the NADH/NAD^+^ ratio which subsequently inhibits the fatty acid β-oxidation pathway in the liver. This inhibition leads to triglyceride accumulation in hepatocytes, resulting in hepatic steatosis and promoting inflammation [52].

#### 2.2.3. Carbohydrate Metabolism

Several pieces of evidence suggest that ethanol assumption may also affect carbohydrate metabolism by generating glycemic excursions depending on whether hepatic glycogen stores are adequate. Indeed, subjects acutely exposed to alcohol showed an increase in glucose intolerance when fasted 12 h previously [53]. In addition, under acute ethanol exposure, a reduction in peripheral glucose utilization occurs, thus inducing hyperglycemia. Another mechanism contributing to hyperglycemia is the increased glycogenolysis, as acute ethanol administration enhances liver phosphorylase activity, rapidly depleting liver glycogen [54,55]. Additionally, ethanol inhibits glycolysis, thus lowering ATP production, thereby increasing the availability of ADP, which stimulates mitochondrial respiration. This process enhances the reoxidation of NADH to NAD^+^, facilitating ethanol oxidation by ADH [56].

In vivo studies have elucidated that the detrimental effects of alcohol consumption on glucose metabolism are enhanced when its administration is combined with a high-calorie diet. As a matter of fact, mice exposed to an intermittent feeding with a high-fat diet and limited access to alcohol exhibited hyperglycemia, insulin resistance, and glucose intolerance, suggesting that binge-eating-like behaviors may transfer to binge-drinking behaviors [57].

Hypoglycemia occurs when ethanol is administered to individuals during the fasting state, principally due to the suppression of hepatic gluconeogenesis. This inhibition is linked to an increased NADH/NAD^+^ ratio during ethanol metabolism, which in turn reduces the conversion of lactate to pyruvate. This reduction limits the formation of phosphoenolpyruvate from pyruvate via oxaloacetate, a rate-limiting step in gluconeogenesis. In addition, the ethanol-induced inhibition of hepatic gluconeogenesis also occurs via the inhibition of CREB recruitment and its coactivator CRTC2 to gluconeogenic promoters through the upregulation of ATF3, a transcriptional repressor that binds to cAMP-responsive elements, thus downregulating gluconeogenic genes [58].

Recently, Chiho Oba-Yamamoto et al. observed that the risk of hypoglycemia increased when the intake of alcohol and glucose was combined as compared to the administration of glucose alone [59].

#### 2.2.4. Protein Metabolism

Acute and chronic ethanol ingestion is also known to promote abnormalities in protein homeostasis and an increase in autophagy, thus compromising muscle mass and favoring the development of secondary sarcopenia [60].

As regards protein synthesis, multiple signals are involved in the regulation of this process integrated by mTOR (mammalian/mechanistic target of rapamycin), whose key member, mTORC1, is known to phosphorylate and activate the main substrates, ribosomal protein S6 kinase (S6K1) and eukaryotic initiation factor 4E binding protein-1 (4E-BP1), which in turn promote mRNA translation via the induction of downstream effectors [61]. When alcohol is chronically consumed, an impaired kinase activity of mTORC1 and suppressed phosphorylation of S6K1 and 4E-BP1 are observed which further prevent the formation of the preinitiation complex for mRNA translation [62,63]. Concerning protein degradation, most of the evidence reported shows that acute alcohol intoxication and chronic alcohol consumption do not accelerate global protein breakdown, probably due to the inability to quantify in vivo muscle proteolysis. Proteolysis is known to be regulated by the ubiquitin–proteasome pathway (UPP) and the autophagic–lysosomal system. In mature rats, acute alcohol exposure reduced the activity of the cytoplasmic proteases alanyl aminopeptidase, arginyl aminopeptidase, and leucyl aminopeptidase 2.5 h after its administration [64]. Conversely, these changes were transient when observed in muscle from chronic alcohol-fed rats where specific protease activity did not differ from that of pair-fed control rats [65]. 

Alcohol-fed mice also showed an increase in autophagy as revealed by the upregulation of LC3B-II, higher mRNA expression of the autophagy-related gene Atg7, and increased BECN1/Beclin1 protein expression [66]. Similar results were also found in alcoholic patients with cirrhosis where the upregulation of LC3B is also accompanied by a sustained reduction in muscle mass [67]. Likewise, hepatic protein content was also altered under alcohol overload in terms of reduction in the synthesis and secretion of the main proteins regulating whole body homeostasis and host defense. Alcohol-induced suppression of liver protein secretion causes an accumulation of export proteins like albumin and transferrin [68]. Notably, proteomic analysis reports that mice known to mimic hepatitis and cirrhosis conditions due to a genetic deletion of ADH, when chronically exposed to alcohol, showed a downregulation of proteins involved in ATP synthesis, carbohydrate metabolism, and lipid transport, as well as reduced expression of apolipoproteins in the plasma [69]. Taken together, these data demonstrate that alcohol abuse shows a dangerous effect on both liver and muscle function, thus favoring a reduction in protein content and the development of different consequences (i.e., altered carbohydrate metabolism and alcoholic myopathy).

#### 2.2.5. Apoptosis

In addition to liver metabolism impairment, alcohol abuse activates many mechanisms altering liver cell homeostasis, facilitating hepatocyte death, resulting in the following progression of alcoholic liver disease until the development of hepatocellular carcinoma.

Chronic alcohol exposure induces multiple forms of cellular stress, such as ROS and toxic metabolites production, able to activate both extrinsic and intrinsic pathways of apoptosis. Additionally, inflammation and immune responses enhance liver cell death together with cytokines and mitochondrial dysfunction, contributing to the cell damage. These processes disrupt liver architecture and function, accelerating the progression of ALD from fatty liver to end-stage liver damage [70].

Several studies describe the toxic effects of alcohol end products on hepatocytes homeostasis. For instance, when accumulated within hepatocytes, acetaldehyde increases tumor necrosis factor (TNF)-mediated apoptosis in a p38 MAPK-dependent manner [71], binds to host proteins, triggers endoplasmic reticulum stress, and forms neoantigens [72].

Furthermore, during alcohol overconsumption, enteric eubiosis and gut barrier permeability are strongly compromised, thus leading to an increased release of LPS into portal circulation. LPS further activates both resident macrophages and infiltrating monocytes, thus favoring the release of pro-death mediators such as TNFα [73]. LPS-derived gut dysfunction also promotes the activation of invariant natural killer T (iNKT) cells in mesenteric lymph nodes, facilitating hepatic cell apoptosis.

As a matter of fact, Llopis et al. recently demonstrated that a specific dysbiosis was associated with severity alcoholic hepatitis, since germ-free mice transplanted with the microbiota of ALD patients developed more severe liver inflammation with an increased number of liver T lymphocyte subsets and NKT cells together with an increased number of hepatic necrotic cells [74].

The liver cell fate under alcohol overload is also compromised by TGFβ-dependent fibrotic processes which appear to potentiate the pro-apoptotic effect of ethanol. As revealed by both in vivo and in vitro results, acetaldehyde increases the expression of TβRII in hepatic stellate cells and promotes the release and activation of latent TGF-β1 [75]. In this scenario, alcohol-induced upregulation of the TGFβ-related cascade promotes a pro-apoptotic signature through the activation of SMAD and non-SMAD/AKT signaling [76]. A decrease in cell viability under ethanol exposure was also observed in the L-02 liver cell line where ethanol-induced apoptosis and cell death was accompanied by ROS production via a NOX/JNK/p38 MAPK-dependent mechanism [77].

Liver cell death after ethanol abuse is also achieved by oxidative stress which leads to the release of damage-associated molecular patterns (DAMPs) and gastrointestinal-derived pathogen-associated molecular patterns (PAMPs), principally obtained from a disrupted gut barrier (Figure 1). These factors lead to the activation of many cell death pathways [78].

#### 2.2.6. Mitochondrial Dysfunction

Alcohol misuse is also known to exert detrimental effects on mitochondrial function by interfering in multiple steps of mitochondrial homeostasis, thus impairing cell viability. A growing body of evidence reported that increased in megamitochondria formation and impaired mitophagy in hepatocytes are the hallmarks of ALD. The generation of megamitochondria is characterized by decreased fission and increased fusion of mitochondria and represents a beneficial adaptive response starting during alcohol-induced hepatotoxicity with the aim to alleviate metabolic stress or to facilitate alcohol metabolism [79].

Nevertheless, when chronically accumulated in liver, megamitochondria may lead to mitochondrial dysfunction resulting in liver injury, an event mediated by the loss of dynamin-1-like protein (DRP1) expression. Indeed, when DRP1 was genetically inactivated in vivo, the development of alcohol-related megamitochondria was exacerbated, resulting in mitochondria maladaptation in terms of a reduction in mitochondria oxygen consumption and hepatic NAD^+^, acylcarnitine, and ketone levels [79,80,81].

In this scenario, for maintaining mitochondrial homeostasis and limiting the accumulation of megamitochondria, defective mitochondria are removed by mitophagy mechanisms [82], whose activation is essential for protecting from ALD. Indeed, Ma et al. have recently reported that alcohol-fed mice with a chronic loss of DRP1 led to mitochondrial maladaptation and impaired mitophagy resulting in dysfunctional innate immune response and aggravated liver injury [81]. Impaired mitophagy was also observed under alcohol overload with a sustained downregulation of mitophagy-related markers (i.e., PINK 1, Parkin), together with decreased ATP levels and disrupted mitochondrial–lysosome fusion in hepatocytes [83].

Furthermore, alcohol also affects mitochondrial respiration as observed in rats fed alcohol both orally and intragastrically. In particular, oral alcohol administration reduced glutamate/malate-, acetaldehyde-, and succinate-driven state III respiration, respiratory control ratio (RCR), and the expression of respiratory complexes (I, III, IV, V) in liver mitochondria. Conversely, intragastric alcohol feeding slightly increased glutamate/malate-driven respiration and significantly enhanced acetaldehyde-driven respiration in liver mitochondria [84,85].

In addition, ethanol-treated animals experience a much more dramatic depression in energy state due to the inhibition of mitochondrial respiration which occurs in a nitric oxide (NO)-dependent fashion as a consequence of the increased activity of NO synthase [34]. In this context, the upregulation of NOX4 seems to contribute to the impairment of respiratory chain complex IV following alcohol withdrawal, since the inhibition of its expression led to restored ROS and ATP levels [86].

Mitochondrial dysfunction under alcohol overload is also characterized by the disruption of the antioxidant system. Indeed, alcohol excess downregulates methionine adenosyltransferase α1 (MATα1), an enzyme that catalyzes the synthesis of S-adenosylmethionine (SAMe), the main methyl donor and a precursor for GSH, the principal antioxidant factor in the liver. In particular, alcohol promotes casein kinase 2-induced phosphorylation of MATα1, thus facilitating interaction with PIN1 which in turn suppresses the mitochondrial translocation of MATα1 (10.1038/s41467-022-28201-2).

Moreover, when mitochondria became dysfunctional, an increased release of mitochondria DNA was observed in hepatocytes from an alcohol-feeding model, which in turn promoted inflammation and apoptosis principally through the cyclic GMP–AMP synthase (cGAS)-interferon regulatory factor 3 (cGAS-IRF3) pathway whose grade of activation is positively related to the severity of ALD [87] (Figure 1).

#### 2.2.7. Inflammation

The occurrence of liver inflammation under alcohol abuse also derives from the disruption of the intestinal barrier and following translocation of lipopolysaccharide (LPS) into systemic circulation. LPS excess affects the liver, thus activating toll-like receptors (TLRs) on Kupffer cells, liver-resident macrophages, resulting in the production of proinflammatory cytokines, including TNFα, interleukins (IL-1β and IL-6), chemokines (IL-8 and CCL2), and ROS [88] leading to the recruitment of blood- and bone-marrow-derived monocytes and neutrophils into the liver (Figure 1).

During alcohol intoxication, damaged hepatocytes also enhance liver inflammation by releasing macrophage migration inhibitory factor (MIF), a multifunctional chemokine promoting the progression of ALD [89]. In fact, when MIF was genetically depleted in vivo, an attenuation of TNFα release and F4/80+ macrophages’ infiltration within the liver was observed after chronic alcohol consumption [89]. As previously discussed, alcohol-induced oxidative stress also generates two types of molecules, PAMPs and DAMPs, that beyond activating pro-apoptotic pathways are also known to trigger inflammatory cascades by recognizing pattern-recognition receptors, such as TLRs. PAMPs, primarily through TLR4, activate NF-κB, leading to the release of CC-chemokine ligand 2 (CCL2), IL-8, and IL-1β, thus promoting the liver infiltration of macrophages and neutrophils. Also, IL-1β binds TLR4, thus activating B cells through the NF-kB pathway. The active form of IL-1β is generated only after the activation of a complex known as the inflammasome, a multiprotein complex involved in the innate immune response via activation of caspase-1 [90]. Moreover, IL-1β sustains liver inflammation and impairs hepatocyte regeneration even after cessation of ethanol exposure because it acts in an autocrine manner by increasing its own concentration and promotes the release of TNF which in turn enhances the sensitivity of hepatocytes to apoptotic signals [91].

The activation of inflammasomes was also evident in both humans and mice with severe alcohol-associated hepatitis, which showed a significantly increased activation of the nucleotide-binding oligomerization domain-like receptor family, pyrin domain containing 3 (NLRP3) inflammasome in the liver after alcohol intake. In addition, under an alcohol binge, the NLRP3 inflammasome activation is also followed by an increase in circulating extracellular apoptosis-associated speck-like protein containing a caspase recruitment domain (ex-ASC) specks and hepatic ASC aggregates which in turn trigger IL-1β release in alcohol-naïve monocytes in mouse models, a mechanism prevented by the NLRP3 inhibitor, thus indicating that these factors are involved in the propagation of systemic and liver inflammation in AH [92].

In the case of prolonged alcohol use, inflammation-mediated damage is repaired by hepatic stellate cells which are primed by nearby Kupffer cells to activate the fibrosis process [93]. Particularly, hepatic stellate cells together with portal fibroblasts produce an extracellular matrix leading to fibrotic tissue deposition [93]. In this scenario, natural killer (NK) cells may abrogate this process by inducing hepatic stellate cell death, but alcohol can directly inhibit NK cell activity, further supporting fibrosis progression [94].

#### 2.2.8. Sex Hormones and Sex Hormone-Binding Globulin

Alcohol intoxication leads to deleterious effects including hormonal disturbances, the extent of which changes according to gender. In fact, compared to men, women still generally consume less alcohol and drink less frequently, even though they show a high risk of developing alcohol-related health problems in the long term, an effect strongly associated with different endogenous concentrations of sex hormones [95]. If on one hand there is an association between steroid sex hormones and gender-related differences in alcohol consumption, on the other hand both acute and chronic alcohol exposure affect the regulation and production of sex hormones, thus inducing the onset of reproductive disorders principally due to the impairment of hypothalamus–pituitary–gonadal (HPG) axis function (i.e., hyperprolactinemia, pituitary hyperplasia, hypogonadism), as we will discuss later [96]. In addition, data from clinical studies have reported that FSH release is more pronounced in males under alcohol craving, as well as there being a positive correlation between progesterone and/or testosterone levels and craving intensity [97]. Importantly, these alcohol dependence-related endocrine fluctuations are also ascribed to the variations in sex hormone-binding globulin (SHBG) production, the main hepatic protein regulating sex hormones’ bioavailability. In this regard, a study on population-based results from a UK Biobank [98] investigated the genetic association between steroid sex hormones and alcohol-related traits, illustrating a positive association between SHBG levels and alcohol intake frequency in both genders [99]. Males with a history of alcohol addiction show an increase in both total testosterone and estrogens, but their bioavailability is lower due to the upregulation of SHBG production, thus suggesting that this binding protein imposes a positive effect on alcohol abuse and vice versa [100]. Nevertheless, increased circulating levels of SHBG occur in alcoholics chronically exposed to ethanol, which further compromises gonad health. Likewise, withdrawal promptly reduces SHBG release, the levels of which achieve reference ranges only after several weeks of abstinence [101].

Notably, the association of SHBG and sexual hormone fluctuations with alcohol addiction is also sustained by heritable traits as reported in a recent GWAS study where SHBG showed a positive genetic correlation with alcohol consumption after adjustment for BMI, thus suggesting that sex-specific heritable predispositions for higher or lower levels of steroid sex hormones and their binding proteins contribute to alcohol-use behaviors and disorders in some contexts [102].

Although the possible mechanism underlying the alterations in SHBG secretion under chronic alcohol ingestion is still uncertain, some evidence has been reported that impaired glycosylation during protein synthesis occurs under alcohol intoxication, resulting in the generation of SHBG isoforms with altered carbohydrate composition [101].

SHBG is also an hepatokine able to exert extra-gonadal effects through the activation of intracellular signaling pathways via binding specific receptor belonging to LDL receptor family (i.e., megalin) expressed in several tissues including cells regulating homeostasis (i.e., neurons, lymphocytes, paraventricular nucleus of the hypothalamus, suprachiasmatic nucleus, adipose stem cells, etc.) [103]. The metabolic role of SHBG has also been described by several observational studies which showed an association between impaired SHBG concentrations with an increased incidence of type 2 diabetes [104,105].

Based on these considerations, chronic alcohol exposure is strongly associated with an increased risk of developing steatohepatitis, type 2 diabetes, and obesity. Multiple pathways are involved in liver damage (mitochondrial dysfunction, steatohepatitis, apoptosis, etc.) and metabolic aggravation (inflammation, oxidative stress) under alcohol overload, each of which may represent a therapeutic target. In this scenario, impaired synthesis and release of SHBG occurs, which in turn enhances gonadal dysfunction and metabolic aggravation. However, the role and mechanisms involved in the dysregulated production of SHBG under alcohol abuse are still unknown.

## 3. The Chemistry of Alcohol: The Effect on Testosterone Production

A potential alcohol-related toxicity effect on testis function is largely hypothesized. Since the 1980s, both acute and chronic alcohol exposure have been shown to induce primary and secondary hypogonadism, reducing hypothalamic gonadotropin-releasing hormone (GnRH) and pituitary luteinizing hormone (LH) production, as well as inhibiting testosterone secretion by the testes [106]. Moreover, in vitro studies have reported ethanol’s ability to promote the aromatization of androgens to estrogens, although this effect in vivo remains to be confirmed [107].

Overall, the exact pathophysiological mechanisms connecting alcohol exposure and steroidogenesis are not yet completely understood. Although many studies have explored the correlation between serum testosterone concentrations and alcohol consumption in humans, a comprehensive view on this topic remains largely lacking, mainly due to the heterogeneity among studies. In order to evaluate the alcohol effect on testicular steroidogenesis, we reported the demonstrations available considering the type of alcohol consumption.

Table 1 summarizes the effect of alcohol consumption on testosterone serum levels as reported in the literature. 

### 3.1. Acute Alcohol Consumption Effects on Testosterone Production

Considering acute alcohol consumption, several studies have reported a reduction in serum testosterone [108,109,110], while others showed a null effect [111]. Moreover, some studies, although the minority, showed testosterone serum levels increasing after acute alcohol consumption [112]. Discrepancies raised by these studies could be explained by several issues that must be carefully considered. First, there is variation in the quantity of alcohol administered in these studies, typically quantified as grams of alcohol per kilograms of body weight, across the studies under consideration. Second, there exists heterogeneity in the time elapsed between alcohol consumption and the assessment of testosterone serum levels which could contribute. Third, the diversity in the types of alcohol utilized (including diluted pure alcohol, spirits, wine, and beer) across studies impacts the generalizability of the findings.

The influence of acute alcohol consumption on testosterone production could be explained by different mechanisms. First, the decrease in testosterone levels associated with acute alcohol consumption could be attributed to a deficiency in several enzymes necessary for steroidogenesis, which are also involved in alcohol and ethanol metabolism. Among the enzymatic processes described above, ADH is expressed in testicular tissues [113]. Thus, alcohol ADH-related metabolism leads to a consumption of the available amount of NAD, which, in turn, is fundamental as a cofactor for some key enzymes in testosterone biosynthesis, such as the 3β-hydroxysteroid dehydrogenase/isomerase complex, 17α-oxidoreductase, and 17α-hydroxyprogesterone aldolase [114]. Second, the potential impact of alcohol on testosterone levels should be assessed at the hepatic level, where acute alcohol consumption may disrupt hepatic metabolism as reported above. Alongside liver damage, these processes could increase the clearance of testosterone [115]. Finally, alcohol intake could have a suppressive effect on the HPG axis, reducing LH and increasing the opioid tone and beta-endorphin [108]. Thus, endogenous opioids increasing could exert a suppressive action on the HPG axis [116,117], although this has not been confirmed by other studies in which subjects showed normal [109] or increased [110] gonadotropin levels following acute alcohol intake.

Studies on male rats revealed other possible mechanisms by which ethanol may impair testosterone secretion, such as the inhibition of NO synthesis [118] and adrenergic neuron activation [119]. Moreover, preclinical studies revealed a direct toxic effect of ethanol (or its metabolite acetaldehyde) on Leydig cells. Alcohol promotes the solubilization of membrane receptors for human chorionic gonadotropin (hCG), resulting in their loss in the surrounding medium, making the pituitary stimulus less effective [120]. Moreover, the integrity and function of Leydig cell mitochondria also appear to be affected by ethanol. However, it should be emphasized that the ethanol concentrations achieved in in vitro models are quite high, and therefore these findings may not significantly transpose to in vivo models.

In conclusion, acute ethanol intake appears to reduce testosterone serum levels by acting on both the gonads and the HPG axis (Figure 2). The extent of this effect is strongly linked to the pattern of intake, such as the mode and amount of alcohol consumed, as well as the clinical state of the subject. Moreover, some of the putative mechanisms revealed by studies in in vitro or in vivo animal models may not be fully transposed to humans, especially when applied to real-life contexts.

### 3.2. Chronic Alcohol Consumption Effect on Testosterone Production

The testosterone serum levels’ trend in chronic alcohol consumption is supported by more solid data. Our recent meta-analysis comprehensively demonstrated that testosterone serum levels’ reduction becomes clinically relevant after chronic alcohol consumption, with around an average 4.86 nmol/L (95%CI: 7.46, 2.26 nmol/L) reduction in subjects exposed to chronic alcohol intake compared to abstaining subjects [121].

Although the ethanol hepatic toxicity is well established in the literature, not all individuals who chronically use/abuse alcohol develop liver damage. In particular, a definitive “safe” maximum quantity of alcohol intake capable of preventing liver damage has not been universally defined [122,123] (Figure 2). Nevertheless, several studies have suggested a greater risk of hormonal imbalance in regular drinkers compared to occasional drinkers. This difference may be attributed to the regenerative capacity of hepatocytes and the lower incidence of nutritional deficiencies observed in the latter group [122,124,125]. Accordingly, although the alcohol-related testosterone decline is comprehensively confirmed, studies available in the literature showed heterogeneous results. Among the 17 studies detected, 10 studies reported lower testosterone serum levels in men chronically using alcohol compared to controls [121]. On the contrary, in 7 studies, testosterone serum levels were equal or slightly higher in alcoholics compared to controls [121].

Considering the mechanisms behind this association, several suggestions should be considered other than reported for acute consumption. First, the lack of compensatory LH increase in alcohol-related testosterone decline [126,127,128,129,130,131] suggests that the adaptation is on the HPG axis, rather than in the testis itself (Figure 2). However, some in vitro studies on Leydig cells showed a reduction in the binding capacity of hCG and presumably LH at the membrane receptors due to ethanol-induced solubilization [120,132]. To further complicate this picture, some authors have suggested that the altered LH response in chronic alcohol users could be related to the frequently encountered hyperprolactinemia in alcohol abusers [133,134,135,136], although the pathogenesis of this phenomenon is not fully understood. Moreover, alcohol-exposed subjects generally show high estrogen serum levels, which could affect the overall feedback on LH (Figure 2). However, although less is known in males about the alcohol effect on estradiol serum levels, it is expected that ethanol could influence aromatase activity; several components of alcoholic beverages, such as procyanidin B and phytoestrogens (i.e., flavones and isoflavones) could bind the active site of aromatase, leading to functional dysregulations [137,138,139]. However, conflicting data regarding estradiol serum levels in chronic alcohol users are available thus far, with some studies showing an elevation [121] and others showing no changes [140]. On the contrary, the increased extra-glandular conversion of weak androgens like androstenedione to less potent estrogens like estrone is strongly detected in chronic alcohol users [141,142] (Figure 2).

Although the evidence in the literature generally agrees on the suppressive effect of chronic alcohol use on serum testosterone levels, it is important to consider the heterogeneity of these studies. Indeed, some studies focused on healthy volunteers, while others included chronic alcoholics. Moreover, the duration of chronic intake varied greatly, ranging from a few weeks to several years (in the case of alcoholics). In this setting, it is also challenging to define the actual alcohol consumption of subjects prior to trial enrollment, as it is mostly based on self-reported questionnaires. Another critical issue concerns the type of alcohol and its administration, as these factors have different impacts on the trend of blood alcohol levels and, consequently, on their action on target organs. Finally, the results obtained in some papers may be affected by the presence of comorbidities or subclinical conditions in the subjects, which may alter the functioning of the HPG axis.

In conclusion, testosterone serum levels’ reduction in chronic alcohol users seems to be independent of the potential alcohol-related liver damage, and abstinence from prolonged and severe alcohol use allows for the spontaneous recovery of gonadal function if irreversible damage to the gonads and liver does not exist [143].

### 3.3. Alcohol Abuse Consequences on Testosterone Production

The definition of abuse alcohol consumption is not univocal. According to the World Health Organization, alcohol abuse is defined as binge drinking (consuming 4 or more drinks for women or 5 or more drinks for men within approximately 2 h, leading to a blood alcohol concentration (BAC) of at least 0.08%) and heavy alcohol consumption (consuming 15 or more drinks per week for men and 8 or more drinks per week for women) [144].

The evaluation of alcohol abuse consequences on testosterone serum levels is even more challenging since men with alcohol-use disorder could also show a range of comorbidities potentially implicated in the genesis of hypogonadism. These patients, generally enrolled in small cohorts, showed varying degrees of liver disease, ranging from minimal alcoholic hepatitis or low-grade portal fibrosis to advanced cirrhosis, potentially affecting testosterone levels and production (Figure 2). Moreover, there was no possibility of directly evaluating gonadal function prior to the onset of alcoholic disease other than through self-report questionnaires. Also, the definition of alcohol-use disorder is variably reported, and these subjects were more prone to further risk behaviors such as substance abuse, with known harmful effects on the reproductive system.

Although potentially linked to testosterone decline in alcohol abusers, the mechanisms described above seem insufficient to explain testosterone changes in abusers. These subjects often exhibit signs of feminization, gynecomastia, palmar erythema, spider angiomas, and female escutcheon [145]. This is the clinical consequence of hyperestrogenism typically detected in alcoholics. Alternative mechanisms advocated involve heightened tissue responsiveness to normal estrogen levels due to the increased expression of estrogen receptors [146] (Figure 2). A reduction in the estrogen clearance rate has also been proposed as an additional mechanism, although available data suggest that estrogen clearance is normal in both alcoholics and subjects with cirrhosis of other etiologies [147]. Moreover, the deleterious effects of alcohol abuse could also result from the dysregulation of other hormonal axes (Figure 2). In alcoholics with cirrhosis, the reduced bioavailability of insulin-like growth factor 1 (IGF1) due to liver disease could partly contribute to increased blood levels of estradiol and estrone and the onset of hypogonadism, given the stimulatory activity of IGF1 on testosterone synthesis [148,149]. Finally, the oxidation of ethanol to acetaldehyde generates a high concentration of free radicals, including the superoxide anion and hydrogen peroxide, which drives oxidative stress affecting cell populations including Leydig cells (Figure 2). Cell membranes are rich in phospholipids, which are highly susceptible to lipid peroxidation. Such damage, repeated over time, promotes the destruction of cell membranes, resulting in the activation of apoptosis and cell necrosis mechanisms [107].

In conclusion, our analysis underscores the dose-dependent impact of alcohol on testosterone levels in males, revealing complex mechanisms involving multiple physiological systems. However, individual responses vary. It is worth noting that the majority of research has focused on adult subjects, leaving a significant gap in understanding the impact of alcohol consumption on adolescents and young adults. Given the increasing prevalence of alcohol consumption among youths and the limited available evidence, further investigation into alcohol’s effects on adolescents during pubertal development is imperative. Enhanced research efforts are needed to fully understand the spectrum of impacts during this critical life stage.

## 4. Distilled Disruption: Unravelling Alcohol’s Effects on Male Reproductive Health

Alcohol intake may adversely affect both the quality and quantity of sperm production, consequently reducing the likelihood of achieving pregnancy. However, the association between alcohol consumption and male fertility has been inadequately evaluated in the literature. Moreover, the association between metabolic syndrome and infertility should be considered similarly [150,151].

Infertility is commonly defined as a couple’s inability to conceive after one year of conception-targeted intercourse without the use of contraception. It is estimated that between 40% and 50% of infertility cases in couples are related to male infertility, and in about 40% of these cases, the condition is considered idiopathic [152]. To date, the main tool used in clinical practice for assessing male fertility remains the semen analysis, although it is known to be a rudimentary investigation burdened with numerous limitations, such as wide intra- and inter-individual variability, operator dependence, and overlapping semen parameters between fertile and infertile men. Here, we present the primary effects of alcohol consumption on male spermatogenesis, taking into account consumption patterns. Table 2 summarizes the effect of alcohol consumption on spermatogenesis as reported in the literature.

### 4.1. Acute Alcohol Consumption Effects on Spermatogenesis

Sperm production could be adversely affected by an increase in oxidative stress and by the amount of ROS at the testicular level. Several conditions can contribute to this phenomenon, from chronic diseases to the environment and lifestyle habits, among which alcohol consumption should be included. However, the real effect of acute alcohol consumption on spermatogenesis is still unclear. At present, only a few clinical trials have investigated the effect of acute alcohol consumption on male fertility, considering the changes occurring directly after acute alcohol ingestion.

While a causal relationship of ethanol on spermatogenesis appears to be evident in studies on animal models, the results from human studies are more unclear and contradictory. Indeed, in animal models, acute alcohol consumption was related to an apoptotic stem cell increase in the seminiferous tubules with a parallel increase in autophagy phenomena in Sertoli cells. Thus, acute alcohol consumption could explain the more frequent occurrence of the Sertoli-cell only syndrome detected in alcohol abusers compared to the general population [153]. However, in humans, the impact of acute alcohol consumption on spermatogenesis should take into account all potential confounding factors related to lifestyle habits. Vieira Silva et al. evaluated 36 young men through semen analysis performed one week before, one week after, and three months after a famous Portuguese festival where for seven days the consumption of alcohol, drugs, and cigarettes increases dramatically [154]. The semen analyses performed after the festival showed a significant decrease in all seminal parameters compared to the baseline, suggesting that the change in lifestyle, with the increased consumption of substances such as alcohol, cigarettes, and drugs, albeit for an isolated period, can alter spermatogenesis. In contrast, large cross-sectional studies did not detect a correlation between alcohol and male fertility [155]. Alongside the different models analyzed, major differences between studies on animal models and those on humans lie in the amounts of alcohol taken into consideration and in the observational study design of human trials. Moreover, it should be noted that these studies have not been able to clearly demonstrate a male fertility impairment due to acute alcohol consumption because spermatogenesis is a lengthy process. Indeed, sperm maturation requires approximately 72 days to complete; thus, a single alcohol exposure may not be sufficient to alter it.

Therefore, it is likely that “sola dosis venenum facit” (the dose makes the poison), and that alcohol consumption, even in acute cases, can affect spermatogenesis and fertility, but at dosages that are difficult to investigate in clinical practice.

### 4.2. Chronic Alcohol Consumption Effect on Spermatogenesis

Data on the effect of chronic alcohol consumption on fertility are challenging to interpret. Indeed, about 20 cross-sectional studies and two meta-analyses on the topic are available. Both meta-analyses demonstrate a negative effect of chronic alcohol consumption on seminal parameters, while the cross-sectional studies are almost evenly split. Eleven of these studies do not show a negative effect of alcohol on seminal health, and only one of them suggests a possible benefit of moderate consumption on male fertility. Conversely, the remaining nine studies conclude that chronic, albeit moderate, alcohol consumption is associated with worse seminal quality and quantity. Overall, a meta-analysis of fifteen cross-sectional studies indicated an inverse correlation between chronic alcohol use and semen quality in 16,395 men, with moderate daily alcohol consumption negatively affecting both semen volume and morphology [156].

The potential beneficial effect of alcohol consumption on sperm production is suggested, advocating the presence of antioxidant compounds in the most common alcoholic beverages such as beer and wine. Indeed, the presence of polyphenols, such as resveratrol or xanthohumol, could offset the toxic effects of moderate alcohol use on the testes. In favor of the potential beneficial effect of moderate alcohol consumption on semen parameters, a prospective cohort study by Karmon et al. showed a positive correlation between male alcohol intake and the probability of achieving a live birth [157]. The authors speculated that this correlation could be related to the improvement in the psychological setting of men consuming alcohol chronically and/or the potential improvement of non-investigated semen factors, such as the sperm DNA fragmentation index (Figure 2). However, in this setting, the potential effect of alcohol consumption on DNA integrity should be considered. Indeed, there are increasing findings that collectively demonstrate the mutagenic potential of alcohol, particularly through mechanisms involving acetaldehyde, oxidative stress, impaired DNA repair, and epigenetic modifications [158,159]. All these mechanisms could affect the potential fecundity of sperm.

These contrasting results could be explained by limitations that must be carefully considered. Most of the works on the topic are retrospective, cross-sectional studies, and the authors based their conclusions on information obtained from databases in which alcohol consumption was not the main outcome. Moreover, these studies are not able to evaluate the presence of possible confounding factors, such as smoking, drug use, and comorbidities, and the alcohol consumption habit is variably reported, with differences also present regarding the definitions of a single alcohol unit, varying from 10 to 13 g. In addition to these limitations, the different results among the studies could also be related to important factors that have not been considered, such as genetic predisposition, diet, or microbiota.

Therefore, the scientific literature struggles to provide a clear answer to the question: “Can moderate, chronic alcohol consumption alter male fertility?” Certainly, there are several variables, not all of which have been properly investigated. It is likely, however, that chronic, albeit moderate, consumption may cause seminal modifications.

### 4.3. Alcohol Abuse Consequence on Spermatogenesis

The definition of alcohol abuse is still challenging, as reported above. Therefore, here we report for clarity those studies evaluating a high daily alcohol consumption, defined as the consumption of more than 50 g daily. Only a few studies are present in the literature on this topic.

In 1978, Kuller et al. conducted a post-mortem study, which, for the first time, highlighted decreased spermatogenesis and testicular damage in a cohort of men classified as heavy drinkers, consuming more than 417 g of alcohol weekly, corresponding to 32 drinks weekly of whiskey, wine or beer [160]. Moreover, they demonstrated a weak correlation between alcohol-related testicular and liver damage, confirming previous results [136] suggesting that alcohol exerts a direct effect on the testis independent of liver injury. Afterwards, in another post-mortem autopsy study, Pajarinen et al. identified more detailed testicular alterations in a small cohort of heavy drinkers, consuming more than 80 g of alcohol daily. Specifically, they demonstrated that in alcoholics, the histological finding of partial or complete spermatogenic arrest is three times more frequent than in non-drinking controls, and that Sertoli cell-only syndrome occurs more than in the general healthy population [161]. These results have been confirmed by Pajarinen and Karhunen, who demonstrated dose-dependent alcohol-related testicular damage [153]. Indeed, when the alcohol consumption was lower than 40 g daily, no testicular damage was described. Otherwise, exceeding the daily dose of 40 g, there was a progressive increase in spermatogenic arrest with a dose-dependent trend.

From a clinical point of view, the first study which evaluated the effects of alcohol consumption on seminal parameters was published in 1985. In this prospective case–control study, the sperm parameters of twenty patients suffering from alcohol-dependence syndrome (consuming at least 165 g of alcohol daily) showed a decreased sperm volume, sperm concentration, and typical forms, although without sperm motility alteration, were compared to ten healthy volunteers [162]. In this analysis, the absent alteration detected in other semen analysis parameters, such as pH and semen viscosity, suggested that alcohol abuse did not affect accessory organs, such as prostate and seminal vesicles. These preliminary data have been confirmed in a larger and more recent prospective, observational, case–control study in which alcoholics consuming more than 60 g of ethanol per day showed an impaired semen volume, sperm count, sperm morphology, and progressive motility compared to healthy controls [163]. As previously reported, semen pH, viscosity, and fructose were not altered in the study group, confirming that the alcohol injury is selective to the testis (Figure 2). However, the spermatogenetic damage caused by alcohol abuse seems to be reversible. Indeed, several case reports and animal studies indicate that spontaneous improvement can occur after 16 weeks of alcohol abolition [164].

Due to methodological limits, it is not possible to identify a causal link between alcohol abuse and sperm production impairment. However, all reported studies show that a condition of chronic alcohol abuse, albeit in the absence of a true limit dose, can result in sufficient testicular damage to alter spermatogenesis through mechanisms that are not entirely clear. Therefore, even in light of the evidence on the possible reversibility of ethanol-induced testicular damage, chronic consumption in large quantities by alcoholics should be discouraged and preferably discontinued if pregnancy is sought.

**Table 1 metabolites-14-00626-t001:** Changes in testosterone serum levels after alcohol consumption as reported in the literature, considering the different habits.

Type of Alcohol Consumption	Testosterone Serum Levels	References
**Acute Consumption**	Reduction	[108,109,110]
	Null Effect	[111]
	Increase	[112]
**Chronic Consumption**	Reduction	[101,121,165,166,167,168,169,170,171,172]
	Null Effect	[173]
	Increase	[97,155,174,175,176,177]
**Abuse**	Reduction	
	Null Effect	[149]
	Increase	

**Table 2 metabolites-14-00626-t002:** Changes in semen analysis parameters after alcohol consumption as reported in the literature, considering the different habits. +: increase; =: null effect; −: decrease; /: not reported.

Type of Alcohol Consumption	Semen Volume	Sperm Concentration	Sperm Motility	Typical Sperm Form	References
**Acute** **Consumption**	−	−	+	−	[154]
**Chronic** **Consumption**	−	=	−/=	−	[156]
	=	=	+	=	[176]
	−	−	+	/	[178]
	−	/	−	−	[179]
	−	−	−	−	[180]
	+	+	+	−	[181]
	−	−	=	−	[169]
	−	+	+	−	[154]
	−	−	+	−	[182]
	−	+	−	+	[183]
	−	−	−	−	[163]
**Abuse**	−	−	/	+	[162]
	−	−	−	−	[163]

## 5. Conclusions

This review highlights the substantial evidence linking alcohol consumption to metabolism and testicular function. Comprehensively, chronic and excessive alcohol consumption could disrupt hormonal regulation along the HPG axis, impair spermatogenesis, and reduce sperm quality. However, the evidence available in the literature is partially conflicting, and several areas warrant further investigation. Thus, while the adverse effects of alcohol on testicular function are increasingly recognized, ongoing research is essential to refine our understanding, guide clinical recommendations, and inform public health strategies.

Alcohol abuse increases gut inflammation and permeability, thus inducing a sustained release of LPS. LPS together with acetaldehyde activates liver-resident macrophages and Kupffer cells to release chemokines and ROS, the key molecules favoring liver inflammation. Alcohol-derived acetaldehyde also enhances the main pathways involved in the regulation of de novo lipid synthesis, mitochondrial dysfunction, and apoptosis in hepatocytes, all events promoting the development of alcoholic fatty liver disease.

## Figures and Tables

**Figure 1 metabolites-14-00626-f001:**
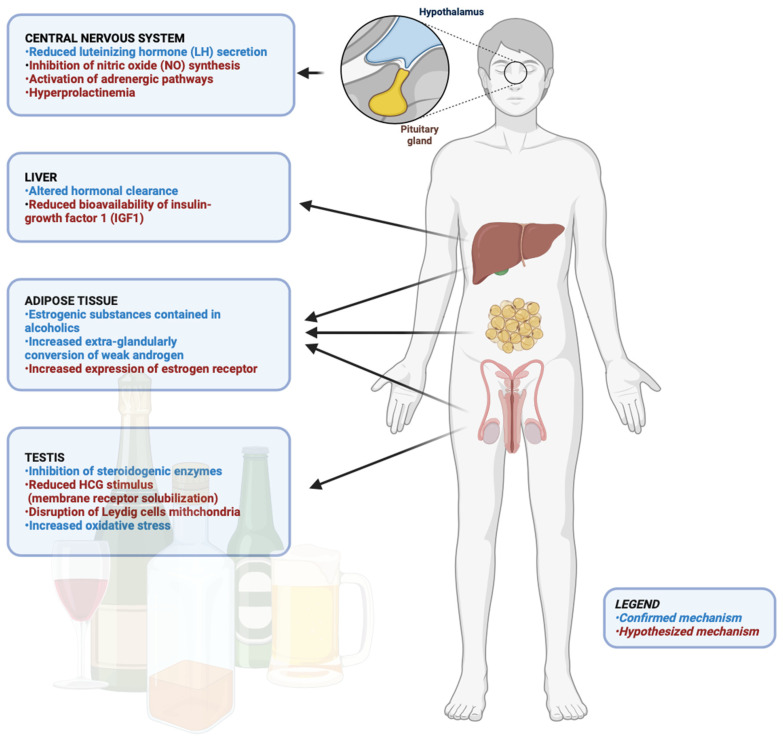
Multiple pathways involved in liver dysfunction under chronic alcohol consumption. ADH = alcohol dehydrogenase; AMPK = 5′ AMP-activated protein kinase; CYP2E1 = P450 2E1 cytochrome; DAMPs = damage-associated molecular patterns; cGAS-IRF3 = cyclic GMP–AMP synthase-interferon regulatory factor 3; PAMPs = gastrointestinal-derived pathogen-associated molecular patterns; PPARα = peroxisome proliferator-activated receptor alpha; LPS = lipopolysaccharide; ROS = reactive oxygen species.

**Figure 2 metabolites-14-00626-f002:**
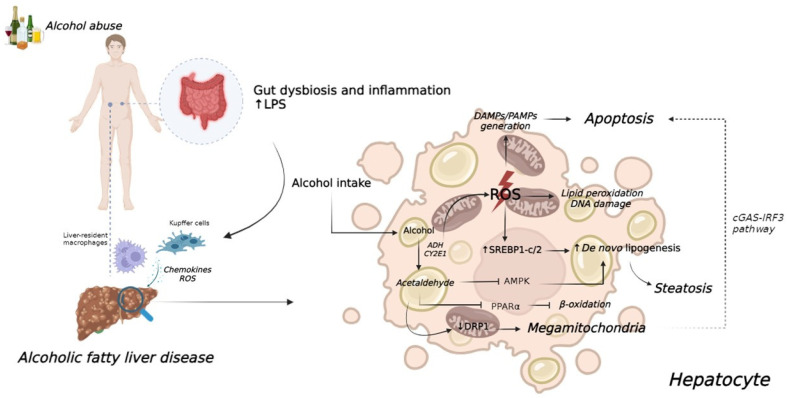
Relationships between alcohol consumption and target tissues/organs, focusing on hormonal effects. Suggested (red) and demonstrated (blue) mechanisms potentially influencing the hypothalamic–pituitary–gonadal axis are reported. HCG = human chorionic gonadotropin.

## Data Availability

The original contributions presented in this study are included in the article.

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
