# Peer review of "Understanding the Role of Alcohol in Metabolic Dysfunction and Male Infertility"

_metabolites, 2024, doi:10.3390/metabo14110626_

Round 1

Reviewer 1 Report

Comments and Suggestions for Authors

In the review entitled Understanding the Role of Alcohol in Metabolic Dysfunction and Infertility by Genchi et al., the authors discuss the role of alcohol in affecting metabolism, followed by a focus on male infertility. The authors have addressed an important topic, as the role of alcohol in male infertility is not well understood. However, several improvements to the review are suggested as mentioned below:

·       Since the authors focus primarily on metabolic dysfunction and male infertility, the title should be modified accordingly.

·       The discussion on alcohol metabolism is very superficial. For a proper understanding of the review, which focuses on the metabolic dysfunction caused by alcohol intake, the authors need to provide an in-depth explanation of the mechanisms involved. This may discuss the mechanism (pathways) of ethanol metabolism, such as the roles of alcohol dehydrogenase (ADH), aldehyde dehydrogenase (ALDH), and the microsomal ethanol oxidizing system (MEOS).

·       Additionally, the products generated, such as acetaldehyde, acetyl-CoA, acetate, and NADH, should be discussed along with how these by-products contribute to oxidative stress and cellular damage. How alcohol effects ATP levels and drives fatty acid synthesis beta oxidation e.t.c. must also be mentioned.

·       The authors should also discuss how alcohol metabolism affects carbohydrate, lipid, and protein metabolism, leading to dysfunctions like insulin resistance and fat accumulation, by describing the mechanisms involved (currently only superficially mentioned in the review).

·       Since the review emphasizes metabolic dysfunction, it would be crucial to dedicate a section to metabolic diseases (in brief) linked to alcohol consumption. For example, brief discussions on conditions like fatty liver disease, alcoholic hepatitis, and the connection between alcohol consumption and diabetes should be included.

·       In other sections under Section 2, such as apoptosis, inflammation, and mitochondrial dysfunction, further clarity is required. Very few studies have been discussed in this review, whereas these is a huge amount of literature available regarding the same.

·       In Sections 3 and 4, the discussion on the relationship between alcohol consumption and male infertility needs more depth. The authors may include a table summarizing the effects of different levels of alcohol consumption (acute vs. chronic, alcohol abuse) on male reproductive health, with separate rows for parameters such as hormone levels, sperm count, motility, morphology, and DNA integrity (based on findings from various researchers).

·       It would also be beneficial to include a figure illustrating the effects of alcohol on the male reproductive system (in-depth, depicting the mechanism) to demonstrate the complex process.

·       The effect of alcohol on genetic mutations as well as the stem cell population in male infertility may also be included as part of the review.

·       In Section 4.1, it is unclear whether the studies discussed under Acute Alcohol Consumption are indeed based on acute exposure. The authors should clarify this. Additionally, they should define what constitutes acute exposure.

·       In Section 4 (Male Reproductive Health), the reference to “remnant studies” on lines 331 and 332 is unclear. If this refers to a specific set of studies, the term needs clarification, or a more accurate description should be provided.

·       A conclusion and future directions should be added at the end of the review, where the authors can summarize the overall findings and discuss the directions and suggestions for future research.

·       The figure legends, particularly for Figures 2 and 3, need to provide sufficient detail to be self-explanatory.

·       The manuscript requires careful editing for grammatical errors and clarity. For instance, the term assumption on line 460 should be corrected to consumption. Similar errors have been observed in other parts of the review, so the authors should thoroughly edit the manuscript to correct such issues.

·       In Section 4.2 (Chronic Alcohol Consumption), the repetition of ideas between the first and second paragraphs should be avoided. It would be better to merge these paragraphs for clarity and to prevent redundancy. Similar observations apply to other parts of the review.

Author Response

Reviewer 1

COMMENT 1

In the review entitled “Understanding the Role of Alcohol in Metabolic Dysfunction and Infertility” by Genchi et al., the authors discuss the role of alcohol in affecting metabolism, followed by a focus on male infertility. The authors have addressed an important topic, as the role of alcohol in male infertility is not well understood. However, several improvements to the review are suggested as mentioned below:

Since the authors focus primarily on metabolic dysfunction and male infertility, the title should be modified accordingly.

ANSWER: Thank you for your comment. We revised the title accordingly.

COMMENT 2

The discussion on alcohol metabolism is very superficial. For a proper understanding of the review, which focuses on the metabolic dysfunction caused by alcohol intake, the authors need to provide an in-depth explanation of the mechanisms involved. This may discuss the mechanism (pathways) of ethanol metabolism, such as the roles of alcohol dehydrogenase (ADH), aldehyde dehydrogenase (ALDH), and the microsomal ethanol oxidizing system (MEOS).

Additionally, the products generated, such as acetaldehyde, acetyl-CoA, acetate, and NADH, should be discussed along with how these by-products contribute to oxidative stress and cellular damage. How alcohol effects ATP levels and drives fatty acid synthesis beta oxidation e.t.c. must also be mentioned.

ANSWER: We appreciate the Reviewer's suggestions; we have deeply discussed the main pathways involved in ethanol metabolism by illustrating both oxidative and non-oxidative pathways (line 63-105). We also described the contribution of main alcohol-related end products on oxidative stress and cellular damage (line 191-231).

COMMENT 3

The authors should also discuss how alcohol metabolism affects carbohydrate, lipid, and protein metabolism, leading to dysfunctions like insulin resistance and fat accumulation, by describing the mechanisms involved (currently only superficially mentioned in the review).

ANSWER: As suggested by the Reviewer, we have better argued the effects of alcohol overconsumption on carbohydrates, lipid and protein metabolism by indicating which are the main metabolic consequences related to these abnormalities (line 250-388).

COMMENT 4

Since the review emphasizes metabolic dysfunction, it would be crucial to dedicate a section to metabolic diseases (in brief) linked to alcohol consumption. For example, brief discussions on conditions like fatty liver disease, alcoholic hepatitis, and the connection between alcohol consumption and diabetes should be included.

ANSWER: We agree with Reviewer’s comments. We inserted a brief discussion regarding the alcohol-related liver diseases including the effects of ethanol consumption on diabetes development (line 130-176).

COMMENT 5

In other sections under Section 2, such as apoptosis, inflammation, and mitochondrial dysfunction, further clarity is required. Very few studies have been discussed in this review, whereas these is a huge amount of literature available regarding the same.

ANSWER: We thank the Reviewer for the suggestions. We included a more extensive description of the effects of alcohol on apoptosis, inflammation and mitochondrial abnormalities by reporting different in vitro and in vivo studies (line 391-615).

COMMENT 6

In Sections 3 and 4, the discussion on the relationship between alcohol consumption and male infertility needs more depth. The authors may include a table summarizing the effects of different levels of alcohol consumption (acute vs. chronic, alcohol abuse) on male reproductive health, with separate rows for parameters such as hormone levels, sperm count, motility, morphology, and DNA integrity (based on findings from various researchers).

ANSWER: Thank you for your comment, we added tables in the section

COMMENT 7

It would also be beneficial to include a figure illustrating the effects of alcohol on the male reproductive system (in-depth, depicting the mechanism) to demonstrate the complex process.

ANSWER: Figure 3 provides, in our view, a sufficiently detailed description of the mechanisms (suggested or demonstrated) linking alcohol with the hypothalamic-pituitary-gonadal axis, while the molecular mechanisms are presented in Figure 1. Combining these aspects into a single figure would limit clarity and usability. However, we have revised the caption for Figure 3 to more accurately reflect its content.

COMMENT 8

The effect of alcohol on genetic mutations as well as the stem cell population in male infertility may also be included as part of the review.

ANSWER: In the literature, there is growing scientific evidence linking alcohol consumption, particularly chronic and excessive intake, to genetic mutations. However, these mechanisms are advocated, generally, to evaluate the oncological risks. We included this concept and its potential implication in male infertility in the specific chapter.

COMMENT 9

In Section 4.1, it is unclear whether the studies discussed under Acute Alcohol Consumption are indeed based on acute exposure. The authors should clarify this. Additionally, they should define what constitutes acute exposure.

ANSWER: The acute alcohol consumption is referred to the changes occurred immediately after the consumption of a large quantity of alcohol within a short period. The Section 4.1 describes the effect of this type of alcohol consumption. We specified this aspect in the manuscript.

COMMENT 10

In Section 4 (Male Reproductive Health), the reference to “remnant studies” on lines 331 and 332 is unclear. If this refers to a specific set of studies, the term needs clarification, or a more accurate description should be provided.

ANSWER: We revised the sentence, referring to the comprehensive demonstration reported in the review cited

COMMENT 11

A conclusion and future directions should be added at the end of the review, where the authors can summarize the overall findings and discuss the directions and suggestions for future research.

ANSWER: Thank you for your suggestion. We added the conclusion section of the review.

COMMENT 12

The figure legends, particularly for Figures 2 and 3, need to provide sufficient detail to be self-explanatory.

ANSWER: We revised the figure legends.

COMMENT 13

The manuscript requires careful editing for grammatical errors and clarity. For instance, the term consumption on line 460 should be corrected to consumption. Similar errors have been observed in other parts of the review, so the authors should thoroughly edit the manuscript to correct such issues.

ANSWER: We revised the manuscript according to your suggestion.

COMMENT 14

In Section 4.2 (Chronic Alcohol Consumption), the repetition of ideas between the first and second paragraphs should be avoided. It would be better to merge these paragraphs for clarity and to prevent redundancy. Similar observations apply to other parts of the review.

ANSWER: We revised the manuscript according to your suggestion.

Reviewer 2 Report

Comments and Suggestions for Authors

The manuscript of Genchi et al., is a review about the negative effects of alcohol consumption on metabolic disturbances and male infertility.

It is a well designed and organized review of the existing literature on the subject. There are some minor comments I would like to address:

1. The review focuses in male infertility and this should be reflected in the title.

2. Authors should provide  the abusive limits of alcohol consumption according to existing studies on the field. 

3. Line 107, 131 etc: " Pathophysiology of alcohol-associated diseases:" should not be repeated and I believe that in should be 2.2.1, 2.2.2 etc 

4. In general, authors should clearly diferentiate between animal and human studies.

5 some minor typos: Line 225: "a history" instead of "an history"; Line 234: "positive genetical correlation"; line 397: "which drives" insted of "which drive"

Author Response

Reviewer 2

The manuscript of Genchi et al., is a review about the negative effects of alcohol consumption on metabolic disturbances and male infertility.

It is a well designed and organized review of the existing literature on the subject. There are some minor comments I would like to address:

  1. The review focuses in male infertility and this should be reflected in the title.

ANSWER: Thank you for your comment. We revised the title accordingly.

  1. Authors should provide the abusive limits of alcohol consumption according to existing studies on the field.

ANSWER: Thank you for your comment. We introduced in chapter 3.3 the definition widely accepted for alcohol abuse.

  1. Line 107, 131 etc: " Pathophysiology of alcohol-associated diseases:" should not be repeated and I believe that in should be 2.2.1, 2.2.2 etc

ANSWER: Thank you for your comment. We revised accordingly.

  1. In general, authors should clearly diferentiate between animal and human studies.

ANSWER: Thank you for your comment. We revised accordingly.

  1. some minor typos: Line 225: "a history" instead of "an history"; Line 234: "positive genetical correlation"; line 397: "which drives" insted of "which drive"

ANSWER: Thank you for your comment. We revised accordingly.

Reviewer 3 Report

Comments and Suggestions for Authors

Dear Authors

Thank you for this very interesting review, which you have done excellently. Considering the close association between alcohol abuse and metabolic syndrome I would only add a brief description of the negative effects of metabolic syndrome itself on male fertility. I leave in evidence a few articles that elaborate on this topic: 

Salvio G, Ciarloni A, Cutini M, Delli Muti N, Finocchi F, Perrone M, Rossi S, Balercia G. Metabolic Syndrome and Male Fertility: Beyond Heart Consequences of a Complex Cardiometabolic Endocrinopathy. Int J Mol Sci. 2022 May 14;23(10):5497. doi: 10.3390/ijms23105497. PMID: 35628307; PMCID: PMC9143238.

Lotti F, Marchiani S, Corona G, Maggi M. Metabolic Syndrome and Reproduction. Int J Mol Sci. 2021 Feb 17;22(4):1988. doi: 10.3390/ijms22041988. PMID: 33671459; PMCID: PMC7922007.

Congratulations again, best regards

Author Response

Reviewer 3

Thank you for this very interesting review, which you have done excellently. Considering the close association between alcohol abuse and metabolic syndrome I would only add a brief description of the negative effects of metabolic syndrome itself on male fertility. I leave in evidence a few articles that elaborate on this topic:

Salvio G, Ciarloni A, Cutini M, Delli Muti N, Finocchi F, Perrone M, Rossi S, Balercia G. Metabolic Syndrome and Male Fertility: Beyond Heart Consequences of a Complex Cardiometabolic Endocrinopathy. Int J Mol Sci. 2022 May 14;23(10):5497. doi: 10.3390/ijms23105497. PMID: 35628307; PMCID: PMC9143238.

Lotti F, Marchiani S, Corona G, Maggi M. Metabolic Syndrome and Reproduction. Int J Mol Sci. 2021 Feb 17;22(4):1988. doi: 10.3390/ijms22041988. PMID: 33671459; PMCID: PMC7922007.

Congratulations again, best regards

ANSWER: Thank you for your comment. We revised accordingly.

Round 2

Reviewer 1 Report

Comments and Suggestions for Authors

The authors have improved the manuscript significantly. They have addressed all the concerns raised by me.